# Phenotype-driven precision oncology as a guide for clinical decisions one patient at a time

Shumei Chia[1], Joo-Leng Low[1], Xiaoqian Zhang[1], Xue-Lin Kwang[2], Fui-Teen Chong[2], Ankur Sharma[1], Denis Bertrand[1], Shen Yon Toh[2], Hui-Sun Leong[2], Matan T. Thangavelu[1], Jacqueline S.G. Hwang[3], Kok-Hing Lim[3], Thakshayeni Skanthakumar [2], Hiang-Khoon Tan[3], Yan Su[1], Siang Hui Choo[1], Hannes Hentze[4], Iain B.H. Tan[1,2], Alexander Lezhava[1], Patrick Tan[1], Daniel S.W. Tan[2], Giridharan Periyasamy[1], Judice L.Y. Koh[1], N. Gopalakrishna Iyer[2] & Ramanuj DasGupta[1]

Genomics-driven cancer therapeutics has gained prominence in personalized cancer treatment. However, its utility in indications lacking biomarker-driven treatment strategies remains limited. Here we present a "phenotype-driven precision-oncology" approach, based on the notion that biological response to perturbations, chemical or genetic, in ex vivo patient-individualized models can serve as predictive biomarkers for therapeutic response in the clinic. We generated a library of "screenable" patient-derived primary cultures (PDCs) for head and neck squamous cell carcinomas that reproducibly predicted treatment response in matched patient-derived-xenograft models. Importantly, PDCs could guide clinical practice and predict tumour progression in two $n = 1$ co-clinical trials. Comprehensive "-omics" interrogation of PDCs derived from one of these models revealed YAP1 as a putative biomarker for treatment response and survival in ~24% of oral squamous cell carcinoma. We envision that scaling of the proposed PDC approach could uncover biomarkers for therapeutic stratification and guide real-time therapeutic decisions in the future.

[1] Genome Institute of Singapore, A*STAR, Cancer Therapeutics & Stratified Oncology, PerkinElmer-GIS Centre for Precision Oncology, 60 Biopolis Street, #02-01 Genome, Singapore 138672, Singapore. [2] National Cancer Centre Singapore, Cancer Therapeutics Research Laboratory, 11 Hospital Drive, Singapore 169610, Singapore. [3] Department of Anatomical Pathology, Singapore General Hospital, Outram Road, Singapore 169608, Singapore. [4] Biological Resource Centre (BRC), A*STAR, 20 Biopolis Way, #07-01 Centros, Singapore 138668, Singapore. Shumei Chia, Joo-Leng Low and Xiaoqian Zhang contributed equally to this work.  Correspondence and requests for materials should be addressed to N.G I. (email: gopaliyer@nccs.com.sg) or to R.D. (email: dasguptar@gis.a-star.edu.sg)

Precision medicine, which has been largely genomics driven[1–4], is defined by being able to treat "the right patient, with the right drug, at the right time"[5, 6]. Given that tumour heterogeneity is underrepresented in many preclinical cell line models, it is not surprising that most drugs fail to demonstrate the efficacy necessary for clinical application[7–9]. Importantly, the use of conventional pre-established in vivo surrogate models, such as patient-derived xenografts (PDX), often lack the desired effect on management as they may not be available in a clinically relevant time frame. Apart from being cost prohibitive, the utility of such PDX models are also severely limited by the fact that they cannot be used to interrogate therapeutics or genetic vulnerabilities in high-throughput screen (HTS) format[10]. Therefore for precision oncology to be successful, novel disruptive technologies are needed that can generate "HTS-amenable" real- or accelerated-time patient-specific ex vivo models, and can also feed information back to the clinic in a relevant time frame. To fulfil the unmet need for novel and predictive approaches to allay treatment failure from the outset, and/or treat recurrences[11, 12], here we present an approach for treatment individualization which is based on phenotypic screening in patient-derived models. We report the generation of a live biobank of patient-derived primary cultures (PDCs) representing primary, metastatic or recurrent tumours obtained from patients with high-risk head and neck squamous cell carcinomas (HNSCCs). HTS-based phenotypic screening of the PDCs, and subsequent validation in matched PDX models for therapeutic vulnerabilities revealed previously unexpected treatment strategies that could potentially allow drug repurposing, and the discovery of novel biomarkers for treatment outcome and resistance. Strikingly, the use of phenotypic screening was translated to the clinic for two different patients, both of whom elicited robust favourable responses as part of co-clinical trials. Additionally, "pan-omics" interrogation of patient-derived models suggested that they largely retain the molecular signatures and phenotypic characteristics of the parental tumour, thereby serving as robust assay/discovery platforms. Altogether results from this study support the hypothesis that treatment response (phenotype) in real-time patient-derived models may serve as the best biomarker for response in the clinic, especially for indications lacking biomarker-guided treatment[13].

## Results

**PDC models predict therapeutic vulnerabilities in vivo.** We established a pipeline to generate PDC and their matched PDX models (Fig. 1a) from pairs of primary and metastatic or recurrent tumours. To date, the success rate of generating PDXs is ~58.1% (25 SCC PDX models out of 43 tumour samples from 24 patients), a number with associated PDCs (Table 1). Notably, among these 24 patients, at least one PDX model was successfully established for each of the 20 patients (~83.3%) (Table 1). Short tandem-repeat (STR) profiling (Supplementary Fig. 1a), gene expression (Fig. 1b) and targeted sequencing of ~763 cancer-related genes (POLARIS Xplora panel; Supplementary Fig. 1b and Supplementary Tables 1 and 2) demonstrated that while genetic alterations and expression profiles were distinct for different patients/models (HN137, HN148 and HN124), each model was highly representative of the original tumour. Three-way pair-wise comparison of the patient-tumour, PDX and source-matched PDCs from primary (HN137-Pri) or from lymph node metastasis (HN137-Met, HN124-Met and HN148-Met) showed high correlation coefficient ($R = $ ~0.9) with minimal divergence in gene expression (Supplementary Fig. 1c) or genetic alterations (Supplementary Fig. 1a).

PDCs derived from seven patients (HN120, HN137, HN148, HN159, HN160, HN177 and HN182) were subjected to high-throughput small molecule screens to uncover pan-cancer as well as patient-specific therapeutic vulnerabilities (Supplementary Fig. 1d). Five of the seven screens were carried out in "real time" (~6 months) either prior to tumour recurrence or in parallel with patient treatment in the clinic. These screens uncovered both shared and patient-specific vulnerabilities (Fig. 1c). For example, HN160 was sensitive to microtubule inhibitors, whereas HN148 showed a particular susceptibility to HDAC inhibitors. It was interesting to note that the patients' mutational profile of cancer-related genes revealed limited predictive potential for therapeutic vulnerabilities (Supplementary Fig. 1b and Supplementary Table 1). This was expected in indications such as HNSCC as it lacks clear biomarker-driven treatment options. Instead, we observed that patient-specific gene expression signatures largely correlated with drug response, thereby suggesting the potential of utilizing these signatures to predict pre-disposition to unique drug sensitivities (Fig. 1d), Finally, the vulnerabilities identified were validated in patient-matched PDX in vivo (Fig. 1e and Supplementary Fig. 1e), demonstrating the translatability of screen hits identified using PDCs.

**Differential gene expression predicts patient-specific response.** A case study of patient HN137: Among the variety of patient models generated, gene expression analysis of the paired HN137-Pri and HN137-Met cells showed the most divergent and interesting differences despite having highly conserved mutational profile of cancer-associated genes (Supplementary Fig. 1a). Therefore, we focussed on the HN137-Pri/Met-isogenic models to investigate whether their unique gene expression signature could confer differential drug response. Pathway enrichment analysis of differentially expressed genes showed an upregulation in cellular processes associated with metastasis in the HN137-Met cells, compared to HN137-Pri. These include the loss of epithelial and cell-junction-related genes (EpCAM, CDH1, CLDN4 and CLDN7) and increase in epithelial-mesenchymal transition signatures (ZEB1/2, VIM, SNAI1 and SERPINE) (Fig. 2a, Supplementary Fig. 2a and Supplementary Tables 3 and 4) ($P$ value <0.0001), which were validated using quantitative-PCR (qPCR) (Supplementary Table 5).

We designed a co-culture screen using HN137-Pri and HN137-Met cells to investigate differential drug response of these two populations. HN137-Pri and HN137-Met cells were labelled with green fluorescent protein (GFP) and tdTomato, respectively, seeded together into each well of a 384-well plate, and screened against a library of anti-cancer compounds and kinase inhibitors, some of which are FDA approved (Fig. 2b and Supplementary Tables 6 and 7). Given the high correlation between the screen replicates (Supplementary Fig. 2c), we identified compounds that displayed pan- and selective toxicity towards HN137-Pri and HN137-Met, respectively (Fig. 2c and Supplementary Fig. 2d). Hierarchical clustering of top-hit compounds with putative targets revealed that HN137-Pri cells were more sensitive to EGFR-targeting tyrosine kinase inhibitors (TKI), whereas HN137-Met were highly sensitive to YM155, an inhibitor of pro-survival genes, including survivin (BIRC5)[14–17] (Supplementary Fig. 2e). These were validated by standard IC$_{50}$ analysis (Fig. 2d), where HN137-Pri showed ~80-fold higher sensitivity to gefitinib compared to HN137-Met (IC$_{50}$ 0.183 vs. 16 µM), while HN137-Met showed 10-fold higher sensitivity to YM155 compared to HN137-Pri (IC$_{50}$ 14 vs. 140 nM). Importantly, validation in PDX (Supplementary Fig. 2f) showed concordance with in vitro data: HN137-Pri-PDX showed preferential sensitivity towards gefitinib as compared to HN137-Met-PDX, especially at earlier time points (days 2-6: $P$ value <0.05) (Fig. 2e), while

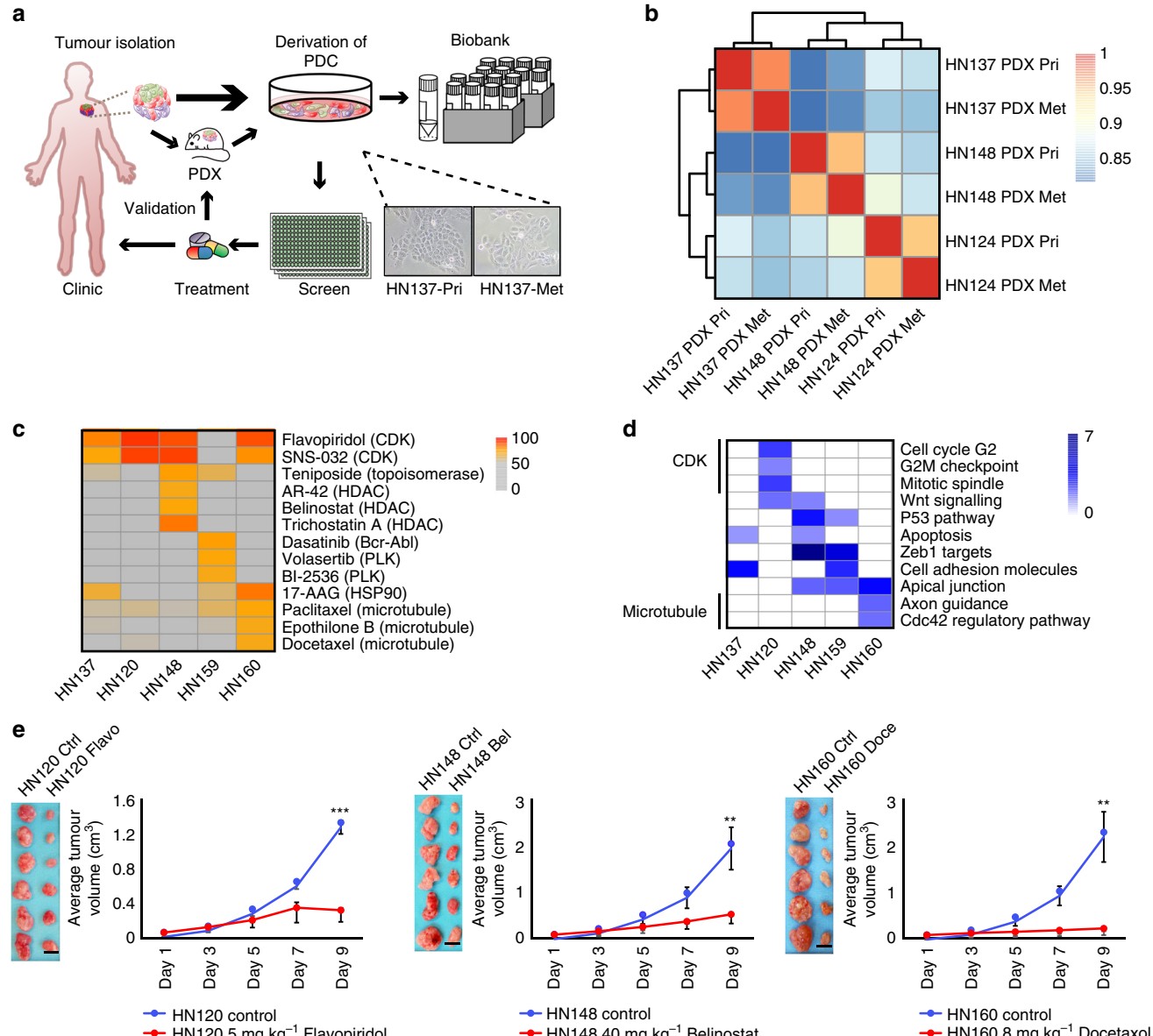

**Fig. 1** Establishing a library of screenable in vitro and in vivo patient-specific squamous cell carcinoma (SCC) models for identification of therapeutic vulnerabilities. **a** Schematic representation of the pipeline for the generation of patient-personalized in vitro and in vivo models that can serve as a screening platform for uncovering therapeutic vulnerabilities. Resected tumours from human patients were grafted into NSG mice for the establishment of in vivo patient-derived xenograft (PDX) models, and for expansion of tumour material. Patient-derived primary culture (PDC) models were also derived and screened against small molecule libraries for identification of patient-specific therapeutics. Sections of patient tumours, as well as freeze-viable PDX and PDC models were stored in our biobank. Representative images of HN137-Pri and HN137-Met cultures are shown. **b** Hierarchical clustering of gene expression profiles for HN124, HN137 and HN148 paired primary (Pri) and metastatic (Met) PDXs in duplicates. *Scale bar* denotes Pearson's correlation coefficient $r$ from 0.8 (*blue*) to 1 (*red*). **c** Heat map of selected anti-cancer compounds exhibiting strong inhibition in at least one of the PDC lines. The *scale* represents percentage inhibition of the compounds, with inhibition score <50% shown in *grey*. **d** Selected molecular signatures ($P < 0.05$) of genes that show elevated expressions across the five Met cell lines, some of which appear to be associated with the selective responses of PDC lines to compounds of same target classes. **e** Six independent cohorts of mice ($n = 6$) bearing patient-matched PDX in one flank were treated with vehicle (control), 5 mg kg$^{-1}$ Flavopiridol (HN120), 40 mg kg$^{-1}$ Belinostat (HN148) and 8 mg kg$^{-1}$ Docetaxol (HN160). *Scale bar*, 1 cm. *Error bars* represent mean ± s.e.m. Two-tail Student's $t$ test was carried out between treatment and control groups on day 9 tumour weight. **$P$ value <0.01 and ***$P$ value <0.001

HN137-Met-PDX exhibited greater susceptibility towards YM155 as compared to HN137-Pri (Fig. 2f), with minimal systemic toxicity (Supplementary Fig. 2g).

**PDC/PDX-guided treatment of patients in co-clinical trials.** One of the major objectives of this pipeline is the ability to influence treatment decisions in real time. The patient (HN137) initially presented with stage IV (T4N2bM0) oral squamous cell

carcinoma (OSCC) was treated with surgery and adjuvant chemo (cisplatin-based)-radiation therapy. Six months after treatment, he developed recurrent disease (including dermal and lung metastasis), suggesting that the patient may have developed resistance to cisplatin (Fig. 3a). We developed cisplatin-resistant HN137-Pri and HN137-Met lines concurrently while the patient was undergoing adjuvant therapy. Interestingly, we found that acquisition of cisplatin resistance does not alter its sensitivity to

**Table 1 Patients recruited for this pipeline, patients' tumour staging, sites of tumour resection and availability of matched PDXs and PDCs**

| S/N | Patient ID | Primary site | Met site | Stage | Primary PDX | Met PDX | PDC |
|---|---|---|---|---|---|---|---|
| 1 | HN120 | Tongue | Lymph node | T4N2b | Yes | Yes | Yes |
| 2 | HN124 | Retromolar trigone | Lymph node | T4N1 | Yes | Yes | No |
| 3 | HN127 | Tongue | Lymph node | T2N2b | Yes | Yes | No |
| 4 | HN132 | Tongue | Lymph node | T2N1 | No | No | No |
| 5 | HN137 | Floor of mouth | Lymph node | T2N2c | Yes | Yes | Yes |
| 6 | HN144 | Alveolar ridge | Lymph node | T4N2c | No | Yes | No |
| 7 | HN145 | Retromolar trigone | Lymph node | T4N1 | No | No | No |
| 8 | HN110 | Tongue | NA | rT2 | Yes | NA | No |
| 9 | HN148 | Alveolar ridge | Lymph node | T4N2a | Yes | Yes | Yes |
| 10 | HN150 | Floor of mouth | Lymph node | T2N2a | Yes | No | No |
| 11 | HN154 | Floor of mouth | Lymph node | T4N1 | Yes | No | No |
| 12 | HN155 | Tongue | Lymph node | T2N1 | Yes | No | No |
| 13 | HN156 | Tongue | Lymph node | T2N2c | Yes | No | No |
| 14 | HN158 | Alveolar ridge | Lymph node | T4N2c | No | No | No |
| 15 | HN159 | Tongue | Lymph node | T4N2b | No | Yes | Yes |
| 16 | HN160 | Buccal | Lymph node | T2N2b | No | Yes | Yes |
| 17 | HN164 | Tongue | Lymph node | T2N2a | No | No | ip |
| 18 | HN165 | Buccal | Lymph node | rT4N3 | Yes | No | ip |
| 19 | HN166 | Buccal | Lymph node | T2N1 | No | Yes | ip |
| 20 | HN169 | Tongue | Lymph node | T4N1 | Yes | No | ip |
| 21 | HN176 | Alveolar ridge | Lymph node | T4N1 | Yes | No | ip |
| 22 | HN177 | Esophagus | Lymph node | T3N1 | NA | Yes | Yes |
| 23 | HN173 | Buccal | Neck | rN2b | NA | Yes | ip |
| 24 | HN182 | Tongue | Lymph node | T2N2b | NA | Yes | Yes |

NA, no patient tumour available; ip, PDX generation in progress

gefitinib (Fig. 3b and Supplementary Fig. 3a), suggesting that the recurrent tumour could be treated using gefitinib.

Based on the therapeutic vulnerability identified using PDCs; their subsequent validation in matched PDX models; and an ongoing study exploring a putative biomarker for EGFR-TKI sensitivity[18], a co-clinical trial was conducted using gefitinib. The patient was treated with monotherapy of gefitinib (250 mg daily) through the IMPACT-SG protocol. This resulted in remarkably significant regression within 6 weeks of treatment (Fig. 3c), supporting the ability of this pipeline to guide patient-specific management of treatment decisions in real time.

To further bolster this notion of using patient-derived models for phenotype-guided clinical decisions, we tested this paradigm in an unrelated case designated HN177. This was a patient who developed widespread metastasis more than a year after definitive treatment for T3N1 oesophageal adenosquamous cancer. Patient HN177 experienced tumour progression on cytotoxic chemotherapy and was hence switched to erlotinib and olaparib. However, the patient developed adverse side effects from this combination with no clinical benefit as measured by no change in tumour markers (Fig. 3f). Patient-specific models generated at the initial presentation of metastases were available for in vitro (PDC) and in vivo (PDX) testing. These showed that erlotinib was effective against the tumour, while olaparib elicited no response (Fig. 3d, e and Supplementary Fig. 3b). Based on these results (and biomarker analyses mentioned previously), the patient was switched to erlotinib monotherapy. This resulted in effective tumour regression with dramatic reduction in the tumour marker CA19-9 (Fig. 3f). Altogether, these observations demonstrate that how PDCs are able to guide or even alter treatment regimens in real time, as patients undergo therapy in the clinic.

**YAP1—a biomarker for metastasis and gefitinib resistance.** Next, we wondered whether the patient-derived models could be utilized to identify potential biomarkers/mechanisms that may

predict tumour progression under the selection pressure of drugs. For patient HN137, despite the initial remarkable response, there was persistence of minimal residual disease in the lungs and dermis, which started to progress after 6 months of gefitinib monotherapy (Fig. 3g). Intriguingly, our PDC models had predicted that HN137-Met cells could represent an aggressive subclone that is naturally more resistant to gefitinib (Fig. 2d). Therefore, we sought to utilize our HN137-Met PDC models to understand the mechanism of resistance to gefitinib.

To determine the context responsible for vulnerabilities identified, we retroactively analysed available "omics" data focussing on the more aggressive HN137-Met PDCs, which showed particular resistance to gefitinib while being sensitivity to YM155 (Fig. 2d). Gene expression profiling revealed that several IAP-family genes were upregulated in the HN137-Met compared to the primary (Figs. 2a and 4a), suggesting a correlation with YM155 sensitivity. Therefore, we explored mechanisms that could account for this increased expression.

Intriguingly, among the upstream transcriptional regulators of IAPs, Yes-associated protein-1 (YAP1), a known activator of pro-survival genes[19], was found to be markedly overexpressed in HN137-Met (Figs. 2a and 4a). Array-based comparative genomic hybridization (arrayCGH) demonstrated a significant amplification of the YAP1 and BIRC3 locus 11q22 (Fig. 4b), specifically in HN137-Met. To assess the chromatin profile of this region, we used formaldehyde-assisted isolation of regulatory elements coupled with sequencing (FAIRE-seq) to profile the chromatin landscape in an unbiased manner. FAIRE-seq analysis revealed de novo opening (thereby suggesting transcriptional activation) of ~14,000 genomic regions (Supplementary Fig. 4a) in HN137-Met compared to HN137-Pri; specifically, this included the YAP1 locus 11q22, and several YAP1 target genes (e.g., CYR61 and CTGF) (Supplementary Fig. 4b). Western blots and qPCR confirmed these findings (Fig. 4c and Supplementary Fig. 4c, d). Interestingly, lower levels of YAP1 expression was detected in HN137-Pri compared to HN137-Met (Fig. 4c).

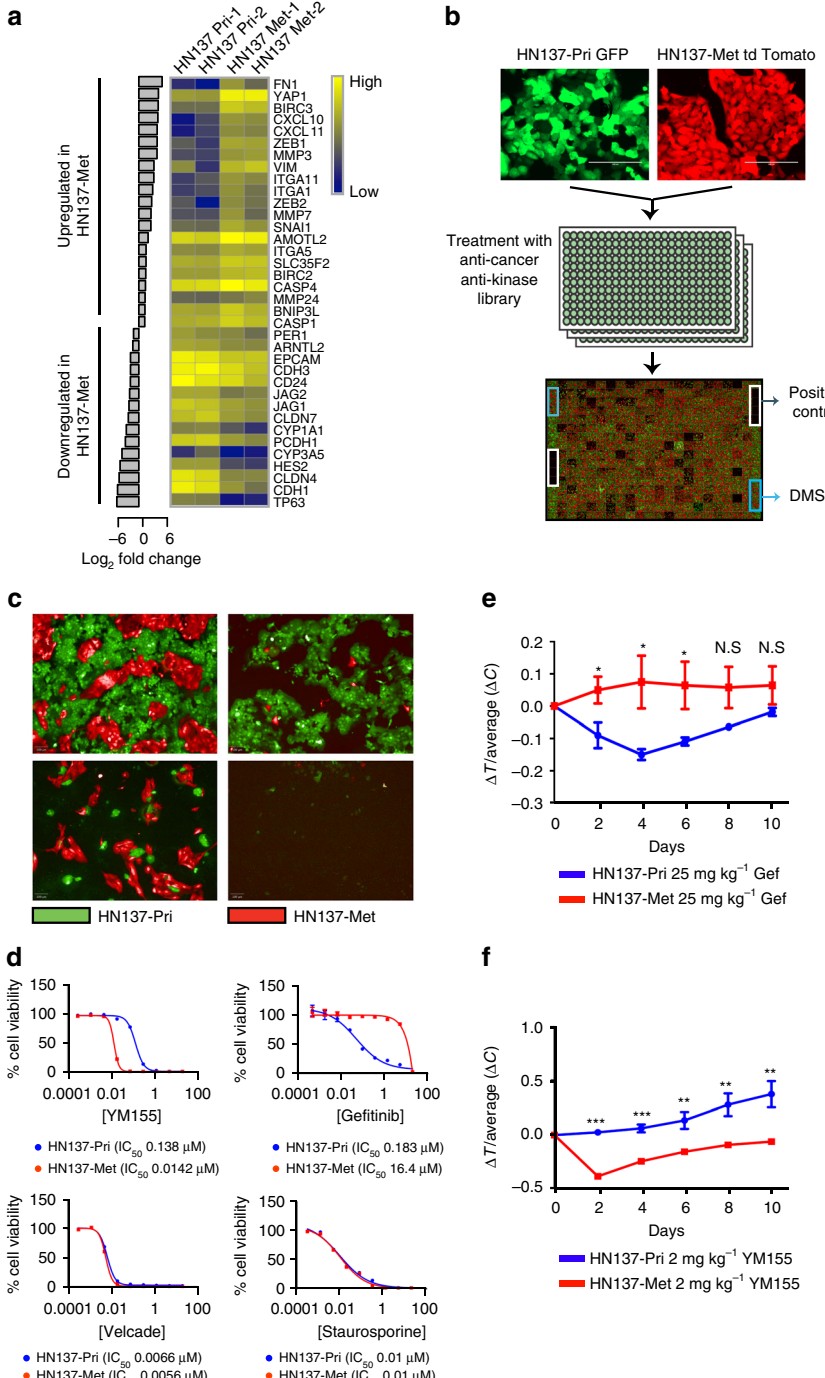

**Fig. 2** Small molecule screen against HN137 PDC revealed patient-specific therapeutic vulnerabilities validates in vivo. **a** Differentially expressed genes (fold-change >2) in HN137-Met vs. HN137-Pri. The heat map represents ($Log_{10}$) normalized gene expression data in duplicates. **b** Schema of the co-culture chemical screen performed in triplicate. **c** Representative fluorescence images of cells treated with compounds specifically targeting either HN137-Pri (*bottom left*), HN137-Met (*top right*) or both (*bottom right*). *Scale bars*, 100 μm. **d** Secondary validation of HN137-Met-specific cytotoxic compound (YM155), HN137-Pri-specific cytotoxic compound (gefitinib) and dual cytotoxic compounds (Velcade and Staurosporine). Experiments were performed at least twice, in triplicates. Cell viability was determined using CellTiter-Glo reagent. Triplicate data, error bars represent mean ± s.d. **e** Four independent cohorts of male mice ($n = 5$), bearing tumours in both flanks from HN137-Pri PDX and HN137-Met PDX, were treated with vehicle (control), or 25 mg kg$^{-1}$ gefitinib (Gef). Note that the HN137-Pri PDX display greater response to gefitinib compared to HN137-Met PDX at earlier days (Days 2-6). *Error bars* represent mean ± s.e.m. Two-tail Student's *t* test was carried out between HN137-Pri and HN137-Met for days 2-10; N.S.: not significant, *$P < 0.05$. **f** Four independent cohorts of male mice ($n = 5$), bearing tumours on both flanks from HN137-Pri PDX and HN137-Met PDX, were treated with 2 mg kg$^{-1}$ of YM155, compared to vehicle (control). A significant anti-tumour effect was observed for YM155 treatment in HN137-Met PDX while HN137-Pri PDX did not display significant sensitivity to YM155. Two-tail Student's *t* test was carried out between HN137-Pri and HN137-Met for days 2-10; *$P < 0.05$, **$P < 0.01$, ***$P < 0.001$

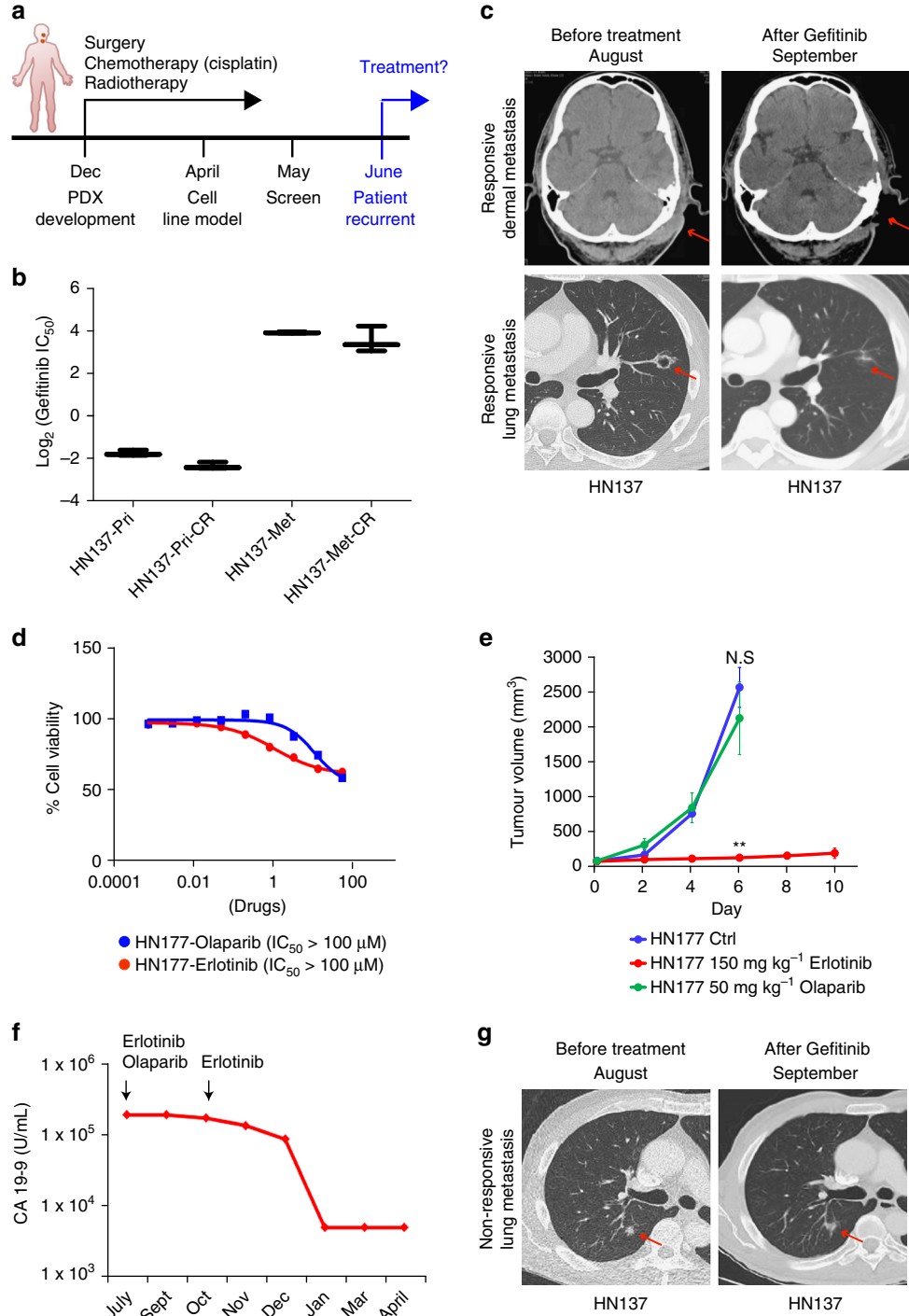

**Fig. 3** PDC/PDX-guided treatment of patients under two independent $n = 1$ co-clinical trials. **a** Timeline for patient HN137 from surgery (December), adjuvant chemo-radiation therapy, until tumour recurrence in June. **b** Graph denoting $Log_2(IC_{50})$ values of HN137-Pri, HN137-Pri cisplatin resistant (CR), HN137-Met, HN137-Met cisplatin resistant (CR) cell lines in the presence of gefitinib. **c** Computed tomography scan (CT-scan) of recurrent responsive metastatic sites (dermal metastasis (*top panel*) and lung metastasis (*bottom panel*)) in HN137 patient, before and after treatment with 250 mg per day of gefitinib. *Arrows* denote sites of tumours before and after treatment. **d** Dose response of HN177-PDC to erlotinib and olaparib. Cell viability was determined using CellTiter-Glo reagent. Triplicate data, *error bars* represent mean ± s.d. **e** Three independent cohorts of mice ($n = 2$ for control and $n = 3$ for treated) bearing HN177-PDX in one flank were treated with vehicle control (Ctrl), 50 mg kg$^{-1}$ olaparib and 150 mg kg$^{-1}$ erlotinib. *Error bars* represent mean ± s.e.m. Two-tail Student's $t$ test was carried out between olaparib and control group (N.S.: not significant) and between erlotinib and control group **$P$ value <0.01. **f** Changes in serum carbohydrate antigen 19-9 (CA 19-9) at initial time of diagnosis and during the course of treatment. **g** CT-scan of recurrent non-responsive lung metastasis in HN137 patient, before and after treatment with 250 mg day$^{-1}$ of gefitinib. *Arrows* denote sites of tumours before and after treatment

Immunohistochemistry (IHC) staining of HN137 primary and metastatic tumour tissue revealed small YAP1-high subpopulations within the HN137 primary tumour (Fig. 4d). Conversely, majority of the HN137-Met tumour displayed uniformly high levels of YAP1. These data suggested that clonal selection and expansion of YAP1-positive tumour cells occurred during metastatic progression, likely by activating pro-survival pathways.

YAP1 signalling enhances cell growth and survival[20]. Therefore to determine if this is necessary for the viability of HN137-Met, we knocked down YAP1 using shRNAs. Downregulation of

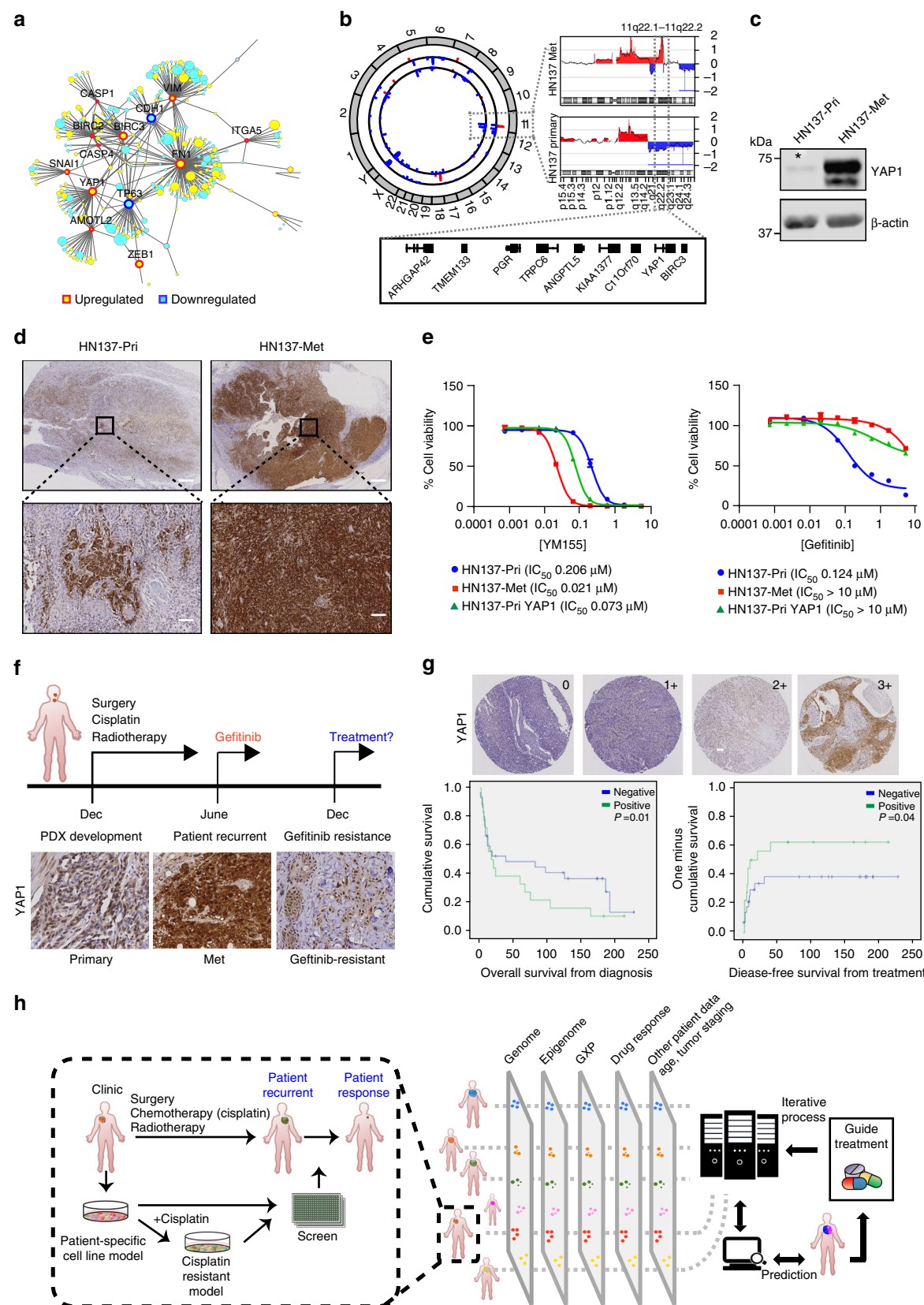

YAP1 resulted in reduced cell proliferation (Supplementary Fig. 4e), reduced expression of pro-survival protein BIRC5 and upregulation of apoptotic markers such as cleaved-caspase3 (Supplementary Fig. 4f), suggesting that YAP1 expression was indeed necessary for survival of HN137-Met tumour cells. To determine the function of YAP1 in modulating drug sensitivity, HN137-Pri cells stably overexpressing YAP1 were generated (Supplementary Fig. 4g), and treated with YM155 and gefitinib. Overexpression of YAP1 in HN137-Pri could sensitize cells to YM155 ($IC_{50}$ 0.07 µM vs. $IC_{50}$ 0.2 µM) while conferring a reciprocal 100-fold resistance to gefitinib ($IC_{50} > 10$ vs. $IC_{50}$ 0.124 µM) compared to control cells (Fig. 4e). Remarkably, YAP1 overexpression had no effect on modulating response to other classes of compounds (Supplementary Fig. 4h), suggesting that its function in conferring sensitivity towards YM155 and EGFR-TKIs is indeed specific. These observations led to the prediction that patient HN137 may relapse with gefitinib-resistant disease, and that the majority of the recurrent tumour would comprise of YAP1-positive cells. This was indeed the case as the patient subsequently developed gefitinib-resistant dermal metastatic lesions ~6 months after initiating gefitinib treatment (Fig. 3d). Importantly, evaluation of a biopsy of the dermal lesion by IHC with YAP1 antibody revealed that the persistent gefitinib-resistant tumour was indeed positive for YAP1 expression (Fig. 4f). Unfortunately, treatment with YM155 (which our data predicts specifically targets this aggressive subclone) (Fig. 2d) was not possible because of the unavailability of this drug for clinical trials.

To investigate the clinical relevance of YAP1 expression levels in OSCC, we performed IHC on tissue microarray (TMA) sections (Supplementary Fig. 4i). TMA sections of 166 tumours[21] were stained and scored for YAP1 expression. Increased expression of YAP1 was observed in 24% ($n = 41$) of patients (Fig. 4g). Survival analyses on high-risk (stages 3 and 4) patients found within our TMA cohort showed that high nuclear YAP1 correlated with poorer overall and disease-free survival ($P = 0.01$ and 0.04, respectively) (Fig. 4g). Altogether, these observations suggest that YAP1 could be used as a diagnostic as well as prognostic biomarker for patient stratification, prediction of gefitinib resistance and metastatic progression.

## Discussion

Rapid advances in sequencing technologies and the ability to more comprehensively interrogate molecular signatures (genomic, transcriptomic, epigenomic and proteomic) have resulted in a strong focus in bringing precision medicine to the clinic. This is especially relevant for cancer research/precision oncology. Over the past several years, genomics-driven technologies have led to major advances in the discovery and mechanistic understanding of oncogenic driver mutations, and pathways that promote cancer progression. However, single cell/multi-sector tumour sequencing studies, and longitudinal tumour progression studies have identified an extraordinary degree of tumour heterogeneity and cellular plasticity, respectively. This may explain how treatment resistance occurs with either conventional chemo- or molecularly targeted therapies. Additionally, even if we were to identify the various driver mutations in individual subclones, not all genomic alterations are actionable, not to mention that developing targeted drugs for every private mutation will take its due course[22]. Therefore, while the genomics-driven strategies will continue to yield insights into tumour biology that may help identify novel therapeutic targets or patient-selection strategies, the clinical benefit, especially as it is currently practiced, may be limited[23, 24]. In this study, we argue that "phenotype-driven precision oncology" using patient-personalized models can provide novel alternative treatment modalities, predictive strategies and help bring "true precision oncology" into the clinic. Additionally, such models can serve as ideal surrogates for retrospective "pan-omics" analyses for the interrogation of mechanisms of treatment failure, resistance and metastasis.

Patient-specific tumour heterogeneity represents one of the greatest challenge undermining fidelity of model generation and therapeutics identification. It appears imperative that we would need to turn towards PDX models for preclinical drug discovery[25, 26] for increased clinical relevance as they are shown by several groups to retain the molecular and architectural features of the original patient tumour[27, 28]. Indeed, a recent publication demonstrated the utility of PDX models in predicting human clinical trial drug response[10, 28]. However, PDX models are not amenable to HTS, thereby greatly limiting the number of clinically available drugs that can be tested for repurposing. Here we propose that a combination of the PDC/PDX system may provide a cost effective pipeline for identification of therapeutics vulnerabilities (or preclinical drug discovery) given the ability to query a larger collection of drugs/combinations. To this end, we showed that therapeutic strategies and prediction of clinical response identified using PDCs could be validated in vivo using patient-matched PDX models (Fig. 1f and Supplementary Fig. 1c), suggesting that this platform can identify hits robustly, with good reproducibility across various models analogous to studies using organoids and more recently, explants[28, 29].

In vitro models however present their own set of limitations. Tumour heterogeneity among various tumour sectors could contribute to inconsistency as the tumour sections used for

**Fig. 4** YAP1 expression as a biomarker for differential therapeutic sensitivity, patient survival, metastatic progression and gefitinib resistance. **a** Protein-protein interaction (PPI) network of differentially expressed genes between HN137-Pri and HN137-Met related to inhibitor-of-apoptosis (IAPs). The width of the edges is proportional to the number of evidences supporting the interaction. Nodes in *yellow* denotes that the genes that are upregulated in HN137-Met, whereas *blue* denotes downregulation. The size of the nodes is proportional to magnitude of fold change. **b** Circos plot depicting global amplification (*red*) and deletion (*blue*) in HN137-Pri (*inner track*) and HN137-Met (*outer track*) genomes. Region of amplification found on chromosome 11 in HN137-Met is shown on the *right*. **c** Western blot for YAP1 expression in HN137-Pri and HN137-Met cells. *Asterisk* denotes band depicting low level of YAP1 expression in HN137-Pri cells. **d** IHC staining of HN137-Pri and HN137-Met tissue specimens for YAP1 expression. *Bottom panels* display higher magnification images of regions marked in *black boxes*. Positive cells were visualized by DAB staining. *Scale bar*, 2 mm (*top panel*), 100 µm (*bottom panel*). **e** Dose-response curve for HN137-Pri, HN137-Met and HN137-Pri cells stably overexpressing YAP1 (HN137-Pri YAP1) to YM155 (*left panel*) and gefitinib (*right panel*). Experiments were performed at least twice, in triplicates. Cell viability was determined using CellTiter-Glo reagent. *Error bars* represent mean ± s.d. **f** Timeline for patient HN137 from surgery and post-operative chemotherapy (Dec), gefitinib treatment (June) till development of gefitinib-resistant dermal metastasis. IHC staining for YAP1 in HN137-Pri, HN137-Met and HN137 gefitinib-resistant dermal biopsy sample (HN137-Gef-R). Positive cells were visualized by DAB staining. *Scale bar*, 100 µm. **g** Representative IHC staining for YAP1 in ($n = 166$) OSCC patient set from Iyer et al.[21]. Graded intensities (0, 1+, 2+ and 3+) of YAP1 nuclear staining shown in the *upper right corner* (*left panel*). *Scale bar*, 100 µm. Kaplan-Meier survival analysis for YAP1 expression (positive: 1+, 2+, 3+; negative: 0) was done. Overall survival from diagnosis (*middle panel*) (Log-rank *P* value = 0.01), and disease-free survival from treatment (*right panel*) (Log-rank *P* value = 0.04) are depicted in number of days. **h** An overview of scaling the proposed "phenotype-driven precision oncology" approach to identify prognostic molecular signatures for treatment response

establishment of models may not be representative of the tumour in its entirety[30]. Once established, clonal selection can occur during the process of model generation, culture adaptation and/or during propagation thereby not retaining the full complexity of the parental tumour[31, 32]. However, our primary screen using HN137-PDCs revealed that a class of molecules targeting EGFR showed similar efficacy against HN137-PDC and PDX models. In another instance, the HN177-PDC also reflected the treatment response of HN177 patient. These results suggest that PDCs can capture to a significant extent, the drug response of the patient tumour in vivo. This is further supported by genomic characterization and gene expression analyses that show key molecular signatures, and oncogenic alterations found in HN137 patient tumours were indeed conserved in PDXs and corresponding PDCs. Similar observations were made with other patient models as well. Therefore, we propose that patient-matched PDCs can potentially serve as predictive models for the identification of alternative therapeutics, in a clinically relevant time frame.

HN137 received cisplatin as post-operative adjuvant chemotherapy. However, tumour recurred, suggesting that cisplatin was not an effective treatment for this patient. Retrospectively, this observation correlated well with our in vitro observation where treatment naive HN137 cells were found to be more resistant to cisplatin as compared to gefitinib (Supplementary Fig. 3a). However, as gefitinib was also effective against recurrent tumour (presumably cisplatin resistant), this observation suggested that gefitinib could be effective against both treatment naive and cisplatin-resistant tumour. Indeed, cisplatin-resistant PDC models developed in vitro were found to be sensitive to gefitinib, correlating well with treatment response in the clinic (Fig. 3b, c).

The development of patient-specific in vitro models that are amenable to genetic manipulation also allowed us to functionally test signatures that could potentially confer differential therapeutic response. YAP1 amplification has been previously reported in OSCCs[33] and implicated in conferring cetuximab resistance[34]. In this study, we demonstrate that overexpresison of YAP1 alone in HN137-Pri cells (to a level similar to those in HN137-Met) was sufficient to confer resistance to EGFR-targeting TKIs, while also increasing their sensitivity to IAP inhibitors. Therefore, high levels of YAP1 which occurs in 24% of OSCCs could serve as potential biomarkers for gefitinib and cisplatin resistance, and these patients can be directed for treatment with IAP inhibitors, such as YM155.

One of the most striking observations made from our models was that they predicted that upregulation or selection of YAP1-positive cells may be responsible for secondary treatment failure upon gefitinib administration. HN137-Met and HN137-Pri overexpressing YAP1 displayed resistance to gefitinib. Indeed, this was demonstrated when patient HN137, who initially responded to gefitinib treatment, developed treatment failure, and as predicted, the resistant tumours were uniformly YAP1 positive. This is particularly significant as it suggests that personalized ex vivo models can predict the trajectory of tumour evolution in a patient, even before it actually occurs in the clinic, thus serving as powerful surrogates to study tumour evolution under different selection pressures.

The proposed phenotypic approach towards precision oncology using patient-matched models aims to complement existing genomics-driven methodologies needed to rapidly and cost effectively translate phenotypic response data into the clinic. While this platform is exciting, it awaits further validation using larger patient cohorts. However, we believe that expanding the repertoire of drug-response phenotypes by running multiple $n = 1$ co-clinical trials will facilitate the development of machine-learning algorithms that may accurately correlate "omics-data" with empirically obtained phenotypes (Fig. 4h). This may result in the identification of novel molecular signatures that could serve as biomarkers that are prognostic of treatment response and progression, especially in indications lacking molecular stratification or in patients without any treatment options. Thus, adopting this complementary approach of pairing genomics with phenotype-driven precision oncology may serve as the best strategy for identifying effective drugs and improving treatment outcomes.

## Methods

**Quantitative real-time reverse transcription PCR.** RNAs were isolated from cells and tissue using QIAzol (Qiagen) prior to purification using RNeasy Mini Kit (Qiagen, cat. no. 74106). RNAs were reverse transcribed using Superscript II reverse transcriptase (Thermo Fisher, cat. no. 18064-071) and quantitative polymerase chain reaction using SYBR Green (KAPA, cat. no. KK4602) were carried out in accordance to the manufacturer's protocol. Transcripts were normalized to *GAPDH* levels. Sequences of primers used for RT-qPCR can be found in Supplementary Table 5.

**Compounds.** Cisplatin was purchased from Tocris (cat. no. 2251), YM155 was purchased from Selleckchem (cat. no. S1130) and gefitinib was purchased from Cayman Chemical (cat. no. 13166). All compounds were dissolved in DMSO (Sigma-Aldrich, cat. no. D8418). Cisplatin was prepared fresh for each treatment.

**Dose-response IC$_{50}$.** Approximately 10,000 cells were seeded per well into a 96-well plate 24 h prior to drug treatment. Drugs were threefold diluted in DMSO and kept at 1% (v/v) across all drug concentrations and control. Each drug concentration was tested in triplicate. The viability of cells were assayed using CellTiter-Glo luminescent cell viability reagent (Promega, cat. no. G7572). The luminescence signals were detected using TECAN Infinite M1000 pro multi-mode plate reader using an integration time of 250 ms. The relative luminescence units from treated wells were normalized against DMSO control wells and expressed as percentage cell viability. IC$_{50}$ values were calculated using GraphPad Prism software.

**Generation and passaging of PDXs.** Tumour samples were obtained from patients post surgery after obtaining informed patient consent in accordance to SingHealth Centralized Institutional Review Board (CIRB: 2014/2093/B). Tumours were minced into ~1 mm$^3$ fragments and suspended in a mixture of 5% Matrigel (Corning, cat. no. 354234) in DMEM/F12 (Thermo Fisher, cat. no. 10565-018). The tumour fragment mixtures were then implanted subcutaneously into the left and right flanks of 5–7 weeks old NSG (NOD.Cg-*Prkdc$^{scid}$ Il2rg$^{tm1Wjl}$*/SzJ) (Jackson Laboratory, stock no. 005557) mice, using 18-gauge needles. Tumours were excised and passaged when they reached 1.5 cm$^3$. For passaging, tissues were cut into small fragment of 1 mm$^3$ prior to resuspension in 20% Matrigel/DMEM/F12 mix, before subcutaneous inoculation of tumour fragments into 5–7 weeks old NSG mice. Protocols for all the animal experiments described were approved by the A*STAR Biological Resource Centre (BRC) Institutional Animal Care and Use Committee (IACUC) under protocol #151065.

**Derivation of PDC cell lines and cell culture.** Tumours were minced prior to enzymatic dissociation using 4 mg mL$^{-1}$ collagenase type IV (Thermo Fisher, cat. no. 17104019) in DMEM/F12, at 37 °C for 2 h. Cells were washed using cyclical treatment of pelleting and resuspension in phosphate-buffered saline (Thermo Fisher, cat. no. 14190235) for three cycles. The final cell suspensions were strained through 70 µm cell strainers (Falcon, cat. no. 352350), prior to pelleting and resuspension in RPMI (Thermo Fisher, cat. no. 61870036), supplemented with 10% foetal bovine serum (Biowest, cat. no. S181B) and 1% penicillin-streptomycin (Thermo Fisher, cat. no. 15140122). Cells were kept in a humidified atmosphere of 5% CO$_2$ at 37 °C. Cell line identity was authenticated by comparing the STR profile (Indexx BioResearch) of each cell line to its original tumour. Cells were routinely screened for mycoplasma contamination using Venor®GEM OneStep mycoplasma detection kit (Minerva Biolabs, cat. no. 11-8100).

**Drug preparation and in vivo treatment.** Gefitinib (Iressa) was prepared by dissolving a 250 mg clinical grade tablet (AstraZeneca) in sterile water containing 0.05% Tween-80 (Sigma-Aldrich, cat. no. P4780) to a concentration of 10 mg mL$^{-1}$ and administered at a dosage of 25 mg kg$^{-1}$ daily via oral gavage. YM155 (Selleckchem, cat. no. S1130) was dissolved in saline to a concentration of 0.5 mg mL$^{-1}$ and administered by intraperitoneal (i.p.) injection, once every 2 days at 2 mg kg$^{-1}$. Flavopiridol (LC Laboratory, cat. no. A-3499) was dissolved in DMSO to a concentration of 200 mg mL$^{-1}$ before diluting to 5 mg mL$^{-1}$ using saline and administered by i.p. injection, once every 2 days at 5 mg kg$^{-1}$. Belinostat (MedChem Express, cat. no. HY-10225) was dissolved in DMSO to a concentration of

100 mg mL$^{-1}$ before diluting to 5 mg mL$^{-1}$ using solvent containing (2% Tween-80 and 1% DMSO in saline), and administered at a dosage of 40 mg kg$^{-1}$ daily via i.p. injection. Docetaxol was prepared in accordance to published formulation[35] and administered by i.p. injection, once every 2 days at 8 mg kg$^{-1}$. Olaparib was solubilized in DMSO and diluted to 5 mg mL$^{-1}$ with saline containing 10% (w/v) 2-hydroxy-propyl-beta-cyclodextrin (Sigma, cat. no. 332607), and administered at 50 mg kg$^{-1}$ daily via i.p. erlotinib was dissolved in 6% captisol (CyDex, Inc., Lenexa, KS) in water, pH 4.5 and administered at 150 mg kg$^{-1}$ daily via i.p. Control groups for all compounds were treated in their corresponding diluent in the absence of compounds.

PDXs were generated by grafting tumours either on both flanks or singly as stated. The length and width of tumours were measured by caliper once every 2 days. Tumour volumes were estimated using the following modified ellipsoidal formula: Tumour volume = 1/2(length × width$^2$). Mice were euthanized when tumours in the control group reaches 2.0 cm$^3$. The weight of tumour was not directly measured, but were estimated using volume where the density of tissue was assumed to be 1 g cm$^{-3}$. The ratio of the change in treated tumour volume ($\Delta T$) to the average change in control tumour volume ($\Delta T$/Average $\Delta C$) at each time point was calculated as follows:

$T$ = Tumour volume of treatment group

$\Delta T$ = Tumour volume of drug-treated group on study day–Tumour volume on initial day of dosing

$C$ = Tumour volume of control group

$\Delta C$ = Tumour volume of control group on study day–Tumour volume on initial day of dosing

Average $\Delta C$ = Average change in tumour volume across the control-treated group.

**Lentiviral and RNAi knockdown of genes**. Knockdown was performed using shRNAs from MISSION® shRNA library (Sigma-Aldrich) cloned into pLKO-1 vector. Overexpression constructs were obtained from the CCSB-Broad Lentiviral expression Library (cat. no. OHS6085, OHS6087, OHS6269, OHS6270 and OHS6271). Plasmid encoding GFP and tdTomato, cloned into pLL3.7 and pLV vector, respectively, was obtained (gift from Tam Wai Leong). Briefly, lentiviruses were packaged in HEK293T cells via co-transfection of plasmid of interest, VSVG and psPAX6 using Lipofectamine 2000 transfection reagent (Thermo Fisher, 11668500). Viruses collected were applied onto the target cells for infection in the presence of 4 µg mL$^{-1}$ polybrene (Santa Cruz Biotechnology, cat. no. sc-134220). The target sequences of the shRNAs are as follow: YAP1 H3 shRNA 5′-GACCA ATAGCTCGAGATCCTTTC-3′; YAP1 H6 shRNA 5′-GCCACCAAGCTAGATAA AGAA-3′. Cells overexpressing proteins of interest were selected using 10 µg mL$^{-1}$ of Blasticidin (Thermo Scientific, cat. no. R21001). GFP and tdTomato-positive cells were FACS sorted.

**Immunoblotting**. Cells were lysed in the presence of RIPA buffer (Thermo Fisher, cat. no. 89900) containing protease inhibitor (Calbiochem, cat. no. 539134) and phosSTOP phosphatase inhibitor cocktail (Roche, cat. no. 04906837001). Lysates were separated on SDS-PAGE gel before blotting onto polyvinylidene difluoride membrane (Millipore, cat. no. IPVH00010). Membranes were blocked using Odyssey blocking buffer (LI-COR, cat. no. 927-40000). Primary antibodies against YAP1 (Abcam, ab52771, 1:2000), baculoviral IAP repeat containing 5 (Santa Cruz Biotechnology, sc-17779, 1:500), β-actin (Santa Cruz Biotechnology, sc-47778, 1:1000), cleaved caspase3 (Cell Signaling Technology, 9664, 1:500), glyceraldehyde 3-phosphate dehydrogenase (Santa Cruz Biotechnology, sc-25778, 1:1000) and V5 (Santa Cruz Biotechnology, sc-81594, 1:1000) were used. Proteins were detected and quantified using secondary antibody (LI-COR, 1:10,000) and imaged on the LI-COR Odyssey scanner. Uncropped scans of blots are supplied in the supplementary information.

**IHC**. Tissue paraffin blocks were sectioned onto polylysine-coated slides and sent to the Institute of Molecular and Cellular Biology (IMCB) Advanced Molecular Pathology Laboratory (AMPL) for IHC. Briefly, the avidin-biotin-peroxidase was performed against YAP1 (Abcam, ab52771, 1:200) for detection. Sections were deparaffinized in xylene and rehydrated through descending percentage of ethanol. Antigen retrieval was performed using citrate buffer (pH 6.1) for 40 min at 65 °C before blocking of endogenous peroxidase activity using 3% solution of hydrogen peroxidase, for 15 min, at room temperature. Slides were blocked using 10% goat serum before application of primary antibody diluted in 10% goat serum. The secondary antibody and HR-peroxidase complex were added and incubated for 30 min at room temperature before washing using TBS-T. The peroxidase activity was visualized by applying diaminobenzidine (DAB) for 5 min at room temperature prior to counterstaining using haemotoxylin. Slides were then dehydrated before mounting and visualization.

**TCGA data**. The gene expression data available for HNSCC was obtained from TCGA (Nature 2015) using cBioPortal for Cancer Genomics (www.cbioportal.org). A Z-score threshold of 2 was used. Genes that co-expressed with YAP1 and their Spearman correlation values were downloaded and plotted using Microsoft Excel.

**Scoring of YAP1 expression in TMAs (tissue microarrays)**. The immunostained slides were scored by two independent pathologists. The specific staining of the YAP1 was observed in five high-power fields (×40). Expression of YAP1 expression was scored as 0, +/−, 1, 2, 3 separately for both nuclear and cytoplasmic staining.

**Generation of Kaplan-Meier plots**. Standard Kaplan-Meier overall survival and disease-free survival plots and the significance of YAP1 expression in patient TMA were generated using SPSS (version 2.0). For the Kaplan-Meier survival analyses, the overall survival and disease-free survival curves are compared using the log-rank test.

**FAIRE-Seq**. The HN137 primary and metastatic cells were cross-linked with 1% formaldehyde for 10 min, which was followed by nuclear isolation and sonication[36]. The FAIRE sites were recovered by phenol:chloroform-isomyl-alcohol extraction of open chromatin, which was quantified and used to generate Illumina next-generation sequencing libraries for analysis with the Illumina HiSeq-Hi-Output. We employed D-filter[37] to call FAIRE peaks in the primary and metastatic cells relative to input control.

**Microarray gene expression**. Flash frozen human tissue and PDX samples were homogenized in the presence of QIAzol (Qiagen, cat. no. 79306) using M-tubes (Miltenyi, cat. no. 130093236) in combination of the gentleMACs dissociator. Cells were grown to ~80% confluency prior to cell lysis using QIAzol. Total RNA was purified from human tissue, PDX and cell line using miRNAeasy kit (QIAGEN, cat. no. 217004). RNAs were sent to genotypic technology for one-color microarray-based gene expression analysis, carried out according to the manufacturer's protocol (Agilent Technologies). Briefly, 500 ng of total RNA were amplified and labelled using Quick-Amp labelling kit one-color (Agilent, cat. no. 5190-0442). The cRNA was hybridized to genotypic's propriety human whole-genome oligo DNA microarray, which includes 50,599 probe set covering 36,337 genes/transcripts (Agilent Human GXP_8X60K, AMADID: 039494).

Signal intensity values from all probes were subtracted with background controls and quantile normalized across samples to correct for batch variation. Correlation between samples were determined from the Pearson product-moment correlation coefficient $r$ of all paired probes, and plotted using R (IDPmisc package).

Probes showing more than twofold change in the expression levels of metastasis and primary samples were considered as differentially regulated, and investigated for overrepresentation of molecular signature gene sets (MSigDB, version 5.1)[38] using Fisher's exact test, followed by multiple testing correction using false discovery rate estimation[39]. Significant gene sets were determined at $P$ value <0.05.

**Comparative MSig analysis across the PDC Met lines**. A reference expression data set was determined from the average normalized expression of each probe in the five PDC Met line. Probes showing more than twofold change in the expression levels of metastasis and reference data set were considered for elevated expressions, and investigated for overrepresentation of molecular signature gene sets using Fisher's exact test. Significant gene sets were determined at $P$ value <0.05.

**Chemical compound library screens and analysis**. Equal number of HN137-Pri GFP and HN137-Met tdTomato cells were seeded for the co-culture screen and treated 24 h post seeding. A total of 4000 cells per well were seeded into 384-well plates (PerkinElmer, cat. no. 6007558). Cells were cultured at 37 °C, 5% CO$_2$ and treated with anti-cancer (Selleckchem, cat. no. L3000) and anti-kinase (Selleckchem, cat. no L1200) small molecule libraries 24 h after cell seeding. Cells were fixed 72 h after treatment using 4% paraformaldehyde for 15 min prior to measurement of fluorescence intensity using the Tecan M1000 and imaging on Operetta High-Content Imaging System (PerkinElmer).

Drug response of various PDC to 317 compounds (Selleckchem anti-cancer compound library) were quantified using the inhibition score (I-score). For each compound c, the I-score was determined as follows:

$$I - score(c) = \frac{\text{Fluorescence signal (c)}}{\text{Median of Fluorescence signal (DMSO)}}$$

where the denominator is the median of the fluorescence emitted in presence of DMSO. Percentage inhibition was determined using (1–I-score) × 100. Comparison between PDC was performed using complete-linkage hierarchical clustering with euclidean distances as distance measure.

GFP (Ex: 488 nm; Em: 507 nm) and RFP (Ex: 554 nm; Em: 581 nm) signals were measured on TECAN Infinite M1000 pro multi-mode plate reader. Drug response to 480 compounds (Selleckchem anti-cancer compound library and kinase inhibitor library) were measured. For each compound c, the Z-score transformation was applied such that $Z = (X_c - \mu)$/s.d., where $X_c$ refers to the fluorescent signals in presence of compound c, $\mu$ refers to the mean I-scores and s.d. is the standard deviation. Compounds that scored $Z < -2.4$ for both RFP and GFP displayed toxicity towards both HN137-Pri and HN137-Met lines. Compounds that scored (RFP $Z < -1$, GFP $Z > -0.5$) were considered as hits

selectively targeting the HN137-Met, whereas compounds that scored (GFP $Z < -1$, GFP $Z > -0.5$) displayed selectivity towards HN137-Pri.

**Comparative genomics hybridization and analysis**. Genomic DNA was isolated from both cell lines and tissue samples as per the manufacturer's protocol (Qiagen, cat. no. 69504). DNA were sent to genotypic technology for comparative genomics hybridization analysis carried out according to their in-house protocol. Briefly, DNAs were label using Agilent Sure Tag Genomic DNA labelling kit (Agilent, cat. no. 5190-3399), a kit that uses random primers and exo-klenow fragment to differentially label genomic DNA samples with different fluorescent label. The labelled DNAs were hybridized onto genotypic proprietary CGH array Human DNA 2X400K chip (Agilent, AMADID: G4124A_068045) using Agilent in situ hybridization kit (cat. no. 5188-6420). After washing, the microarray slides were scanned and the captured images were imported into Agilent cytogenomics 2.9.2.4 software for analysis. At least three consecutive probe sets were used to call a CNV. Abberent intervals were identified using the aberation detection algorithms (ADM-2) with a threshold of 6.0, a minimum absolute average $\log_2$ ratio per region of 0.25 and nesting filter of 100.

The CIRCOS circular genome presentation software[40] was used to plot the histogram of copy number gains and losses (filtered at three and above) detected in HN137-Pri and HN137-Met genomes.

**Protein-protein interaction**. Physical interactions between differentially regulated genes were integrated from BIOGRID release 3.4.137[41]. The network views were constructed using Cytoscape (version 3.2.1), and arranged using spring embedded layout[40]. A proprietary algorithm (J.L.Y. Koh et al., unpublished) was used to determine the sub-network involving key genes that were used as "seeds" to identify first- and second-degree neighbouring nodes. A closed neighbourhood was then identified from these nodes.

**Targeted capture panel**. The capture panel used in this study is an in-house designed panel consisting of 763 genes that are clinically and biologically relevant to cancer. They are derived from a comprehensive literature and database search and include genes involved in key oncogenic signalling pathways, oncogenes, tumour suppressor genes and genes from kinase and chromatin remodeller families. The list of target capture gene panel is provided (Supplementary Table 2). The tumour-normal pairs were enriched with the Xplora capture panel and sequencing was performed on the HiSeq platform.

**Bioinformatics analysis pipeline**. Our bioinformatics pipeline for processing of next-generation sequencing data and variant discovery utilized best practices as described by Genome Analyzer toolkit (GATK) (Broad Institute, https://www.broadinstitute.org/gatk/guide/best-practices.php).

Paired-end sequencing reads were aligned to the human reference genome NCBI GRC Human Build 37 (hg19) using the Burrows-Wheeler Aligner (BWA)-MEM (v0.7.9a)[42] and PCR duplicates were removed using SAMtools (v0.1.8)[43]. Base qualities were recalibrated and realignment around microindels was performed using GATK[44]. Somatic variants within the targeted region were called using MuTect (v1.1.4)[45] and LoFreq (v0.5.0)[46] with default parameters. We filtered any variants called by a single method or with an allele frequency <0.1.

Single nucleotide variations (SNVs) were annotated using Variant Effect Predictor (VEP, v2.8)[47] for hg19. Gene transcript annotation databases (CCDS, RefSeq, Ensembl, UCSC Known Genes) were used to identify transcripts and to determine amino acid changes. Amino acid changes corresponding to SNVs were annotated according to the largest transcript of the gene. SNVs that were present in dbSNP (Build 132) were removed unless they were also present in COSMIC (v52) indicating a previously confirmed somatic event.

**Data availability**. Gene expression and CGH array data that support the findings of this study have been deposited in GEO with the accession code GSE84676. FAIR-seq data that support the findings of this study have been deposited in GEO with the accession code GSE92479. Compound library screen data associated with this study can be accessed via the Centre for High-throughput Phenomics (CHiP-GIS) portal at http://chip.gis.a-star.edu.sg/DOWNLOAD/Chia2017/

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

## Acknowledgements

We would like to extend our gratitude to all patients and families involved in this study. Special thanks to Drs Winston Chan, Tenzin Gocha and Srikala Raghavan for critically reviewing the manuscript. We are thankful for technical support from the following: Tan Cheng Peow Bobby and Yaw Lai Ping for IHC, Shivaji Rikka at the Centre for high-throughput phenomics (CHiP-GIS) and Jonathan Cechetto at the PerkinElmer-GIS Centre for Precision Oncology for cell line-based HTS/HCS, Ho Danliang and Saranya Thangaraju for their bioinformatics support and the A*STAR POLARIS for mutational profiling of cell lines. We thank Dr Tam Wai Leong for sharing the GFP and tdTomato plasmids, and Dr Vibhor Kumar for help with FAIRE-Seq analysis algorithm. We are grateful to the Biological Resource Centre (BRC) at A*STAR, where all the animal work was conducted under IACUC protocol #151065. Co-clinical trial was performed under the IMPACT trial (ClinicalTrials.gov Identifier: NCT02806388. This project was supported by grants from the National Medical Research Council (Singapore) (NMRC/CIRG/1434/2015), Agency for Science, Technology and Research (A*STAR-core funds to RD) and National Cancer Centre Research Fund. N.G.I. is further supported by an NMRC clinician-scientist award (NMRC/CSA/042/2012).

## Author contributions

S.C., N.G.I. and R.D. conceived and designed the study. S.C., J.-L.L., X.Z., X.-L.K., F.-T.C., A.S., S.Y.T., H.-S.L., M.T.T. and G.P. performed experiments. J.L.Y.K. and D.B. performed computational analysis; A.L., P.T., I.B.H.T., Y.S. and S.H.C. coordinated the Polaris Xplora-Panel sequencing. D.S.W.T. conducted the co-clinical trial through IMPACT protocol. H.H. was involved with mouse-PDX studies at the BRC, A*STAR. H.-K.T. was involved in clinical care for patient HN137. J.S.G.H. and K.-H.L. scored the TMAs. T.S. correlated TMA scoring with patient outcome. S.C., R.D. and N.G.I. wrote the manuscript, with extensive input from all authors.

## Additional information

**Competing interests:** The authors declare no competing financial interests.

