## [Peer Review File · Nature Communications]

Reviewers' comments:

Reviewer #1 (Remarks to the Author):

The authors made some substantial efforts to address the reviewers' comments in the revised manuscript. They have performed additional analyses to address the genomic fidelity of the PDXs and PDCs compared to the patient tumors. They also provided additional IHC data to support that overexpression of YAP1 in HN137-Pri cell can promote cellular invasion. Furthermore, they have re-worked the manuscript extensively to be more accurate and precise on the messages.

While most of the points raised by the reviewers have been addressed, there are some major points that need to be further clarified before it is accepted for publication.

- In response to the questions regarding clonal evolution: The authors provided additional data to support their conclusion that there is no significant clonal evolution during PDX passaging and in vitro culture of PDCs. And the response in PDCs and PDXs are consistent between these model systems. Several lines of evidence in the literature (some were cited by reviewers #1 and 2) suggests that more or less PDXs and PDCs undergo clonal selection, however, it seems that the major drivers were retained in the PDXs. This is also reflected in drug responses. As pointed by reviewer #1, a publication by Ricci et al 2014 clearly showed that in two out of 28 cases, the corresponding PDXs responded to a CDDP-based therapy differently when compared to the patients. I would suggest the authors tone down their conclusions a when discussing the clonal changes.
- In response to the questions regarding "preliminary" and "limitations" by both reviewer #1 and #2: The authors have demonstrated the translatability of their approaches with a single case. However, the conclusion cannot be simply extrapolated to other cases, as it is clear that with increased N, there would be cases that the response in PDXs and PDCs may not be consistent with that in patients as has been already shown in Ricci et al 2014 publication. I would suggest that the authors discuss this limitation in the discussion section. The author also offered to include another case which would definitely strengthen their claim.
- In response to Figure 5a regarding the sensitivity of primary/met tumors to gefinitib: the reviewers' concern is not adequately addressed. The difference of responses in primary tumors versus mets is very mild although statistically significant for two time points. The raw data needs to be provided and graphed in the similar way as they did in Figure 1f. $\Delta T/\Delta C$ (tumor volume change in treatment group versus vehicle control group), the standard approach in in vivo pharmacology to assess treatment effect can then be calculated and compared as the authors stated.
- In response to the dosage and exposure as well as PD assessment of YM155: this point is not addressed appropriately. The authors did not observe any animal toxicity and believed that the compound had a good therapeutic window in mice. However, mice often tolerate treatments better than patients, the dose level/exposure in mice may not be achievable in the clinic, and therefore the efficacy observed in mice will have a low likelihood to translate into the clinic. The literature they provided did not have information on what is the clinically relevant dose for YM155. In addition, the authors failed to provide evidence regarding the PD and efficacy correlation. No toxicity observed in the animals does not rule out the potential off-target effects of YM155. YM155 has been reported in the paper cited by the authors to inhibit not only survivin, but also EGFR, MCL-1 etc.
- Page 4 line 74 and figure 1b: the author claimed 90% take rate of generating PDXs. In Figure 1b, primary and metastatic PDXs were established from 5 patients, and 8 additional models were

either established from primary or from mets. The author added up 10 primary PDXs and 8 metastatic PDXs from 16 patients and divided by 20 (16 plus 4 in progress) to get the 90% take rate. This is very misleading. I would suggest that the authors calculate the take rate separately for primary and metastatic tumors. If they prefer to combine them, then it should be 18 PDXs (primary + metastatic)/31 tumors (HN110: mets not available) = 58%, not divided by # of patients.

Response to reviewer comment:

We thank the reviewers for their critical insights and constructive suggestions. We have made every effort to address all of the concerns raised, including the addition of new data, methodologies and a detailed discussion. Most importantly, as noted by the reviewer as a critical requirement to strengthen our claims, we have now included a second case study to bolster the notion that the patient-specific models are indeed predictive and can be used to guide treatment modalities in the clinic. We sincerely hope that the revised manuscript is now deemed appropriate for publication in Nature Communications.

- In response to the questions regarding clonal evolution: The authors provided additional data to support their conclusion that there is no significant clonal evolution during PDX passaging and *in vitro* culture of PDCs. And the response in PDCs and PDXs are consistent between these model systems. Several lines of evidence in the literature (some were cited by reviewers #1 and 2) suggests that more or less PDXs and PDCs undergo clonal selection, however, it seems that the major drivers were retained in the PDXs. This is also reflected in drug responses. As pointed by reviewer #1, a publication by Ricci et al 2014 clearly showed that in two out of 28 cases, the corresponding PDXs responded to a CDDP-based therapy differently when compared to the patients. I would suggest the authors tone down their conclusions a when discussing the clonal changes.

We agree with the reviewer that clonal evolution is possible during the course of PDX generation and cell line derivation. This applies to all cancer models as it is not possible to propagate heterogeneity found in the entire tumor in any culture system, and that populations of more proliferative cells could be enriched. However, our genotyping analysis supports that key oncogenic drivers found in the tumor tissue are conserved in our PDX mice and cell line models, corresponding to the publication 2014. Additionally, we also demonstrate significantly high correlation at the level of gene expression between the patient tumor and *in vitro/in vivo* models. Nonetheless, as requested, we have toned down the discussion and conclusion made on this point in this revised manuscript, and discussed the possibility of clonal selection in the models (line 274-288).

- In response to the questions regarding “preliminary” and “limitations” by both reviewer #1 and #2: The authors have demonstrated the translatability of their approaches with a single case. However, the conclusion cannot be simply extrapolated to other cases, as it is clear that with increased N, there would be cases that the response in PDXs and PDCs may not be consistent with that in patients as has been already shown in Ricci et al 2014 publication. I would suggest that the authors discuss this limitation in the discussion section. The author also offered to include another case which would definitely strengthen their claim.

We thank reviewer for the comment. While we agree that response may not be 100% coherent between patient-derived models and clinical outcome, we want to highlight that at least in the patient-matched *in vivo* PDX models (that are largely considered to be the gold-standard for clinical response^{1,2}), the response was highly concordant with those observed in the PDCs in our system. This is mostly similar to what was observed in the Ricci et al 2014 paper, where the concordance between the models and retrospective clinical outcome data was very high (~93%: 26/28).

In this manuscript, we are however comparing response in models to those in patients prospectively, because of which it is non-trivial to scale in a short time-frame due to regulatory issues. Nonetheless, we have now included a second case to support the reproducibility of our platform, as was suggested by the reviewer and to make this claim more robust. Even so, while we present 2

cases, we agree that the platform still awaits validation with a larger patient cohort. We have hence also included this limitation in the discussion section (line 319-320).

In addition, we have also included in discussion a commentary on how heterogeneity among tumor sector in addition to clonal selection could be potential factors contributing to incoherence in drug response between patient and *in vitro* models (PDX/PDC) established (line 274-288).

- In response to Figure 5a regarding the sensitivity of primary/met tumors to gefitinib: the reviewers' concern is not adequately addressed. The difference of responses in primary tumors versus mets is very mild although statistically significant for two time points. The raw data needs to be provided and graphed in the similar way as they did in Figure 1f. $\Delta T/\Delta C$ (tumor volume change in treatment group versus vehicle control group), the standard approach in *in vivo* pharmacology to assess treatment effect can then be calculated and compared as the authors stated.

We thank the reviewer for the comment. We have revised the figure accordingly (please refer to Fig. 2e-2f, Supplementary Fig. 2f-2g). Specifically, we plotted the ratio of change in tumor volume in treated group to that of control ($\Delta T/\Delta C_{avg}$). As is evident from the results shown in Fig. 2e, the HN137-Pri PDXs responded to Gefitinib treatment more readily at earlier time points, compared to HN137-Met PDXs. Please also refer to detailed protocol in the figure legends and materials & methods. Additionally, we have now made the raw data available, which was used to plot/graph the response in PDX models as requested by the reviewer.

- In response to the dosage and exposure as well as PD assessment of YM155: this point is not addressed appropriately. The authors did not observe any animal toxicity and believed that the compound had a good therapeutic window in mice. However, mice often tolerate treatments better than patients, the dose level/exposure in mice may not be achievable in the clinic, and therefore the efficacy observed in mice will have a low likelihood to translate into the clinic. The literature they provided did not have information on what is the clinically relevant dose for YM155. In addition, the authors failed to provide evidence regarding the PD and efficacy correlation. No toxicity observed in the animals does not rule out the potential off-target effects of YM155. YM155 has been reported in the paper cited by the authors to inhibit not only survivin, but also EGFR, MCL-1 etc.

A Phase 1 clinical trial has been conducted with YM155³. In accordance to this trial, the MTD for YM155 is reported to be 4.8mg/m²/d for human, which is approximately 0.13mg/kg. Mice were treated at 2mg/kg which is equivalent to 0.162mg/kg in human. As the dose administered in mice is very similar to the MTD of YM155 in human, we would like to think that that this treatment regimen is clinically achievable. Calculations for dose conversion between animals and patients were performed as shown below and methodology was adopted from the following published article⁴.

Interchange of unit (mg/kg to mg/m²) of dose of animals or human is carried out using the K_m factor [Table 1] as:

$$\text{mg / m}^2 = K_m \times \text{mg / kg} \quad \text{Eq. (4)}$$

Given that MTD of YM155 is 4.8mg/m² and K_m of human is 37, MTD of YM155 in human is = 4.8/37 = 0.13mg/kg

$$\text{HED (mg / kg)} = \text{Animal does (mg / kg)} \times (\text{Animal } K_m / \text{Human } K_m) \quad \text{Eq. (2)}$$

$$\begin{aligned} \text{Human equivalent dose (HED) mg/kg} &= \text{Animal dose (mg/kg)} * (\text{Animal } K_m / \text{Human } K_m) \\ &= 2\text{mg/kg} * (3/37) \\ &= 0.162\text{mg/kg} \end{aligned}$$

Treatment of mice with 2mg/kg of YM155 is equivalent to 0.162mg/kg in human, which is close to the MTD of YM155 in human

As no patients were treated with YM155, it is difficult to comment on the PD and efficacy correlation of YM155 in the patient. However, the pharmacodynamic properties of YM155 was described in the following phase 1 clinical trial study³.

Most small molecules have multiple targets, however, they are usually lead optimised such that they display increased affinity for the target of interest. The off-target effect of such drugs are typically minimised by identifying the ideal therapeutic window, and administering them at limited or controlled dosage.

We understand that YM155 may have multiple targets as reported in published literature^{5,6} and it is possible that YM155 could be acting through multiple pathway in our cells. However, we think that the downregulation of EGFR (which is one of the reported off target effect of YM155) is not likely to contribute to HN137-Met sensitivity to YM155 as these cells are inherently more resistant to Gefitinib, a small molecule TKI that targets EGFR. However, one should note that this limitation (which is typical of all small molecules), does not alter the conclusion that our platform can be used to identify alternative therapeutics, with beneficial response/outcome in patients.

- Page 4 line 74 and figure 1b: the author claimed 90% take rate of generating PDXs. In Figure 1b, primary and metastatic PDXs were established from 5 patients, and 8 additional models were either established from primary or from mets. The author added up 10 primary PDXs and 8 metastatic PDXs from 16 patients and divided by 20 (16 plus 4 in progress) to get the 90% take rate. This is very misleading. I would suggest that the authors calculate the take rate separately for primary and metastatic tumors. If they prefer to combine them, then it should be 18 PDXs (primary + metastatic)/31 tumors (HN110: mets not available) = 58%, not divided by # of patients.

We thank the reviewer for the comment. We have revised the figure in accordance to the reviewer's comment. Please refer to Fig. 1b and line 74-76 in text.

References:

1. Gao, H., *et al.* High-throughput screening using patient-derived tumor xenografts to predict clinical trial drug response. *Nat Med* **21**, 1318-1325 (2015).
2. Pauli, C., *et al.* Personalized In Vitro and In Vivo Cancer Models to Guide Precision Medicine. *Cancer discovery* **7**, 462-477 (2017).
3. Tolcher, A.W., *et al.* Phase I and pharmacokinetic study of YM155, a small-molecule inhibitor of survivin. *Journal of clinical oncology : official journal of the American Society of Clinical Oncology* **26**, 5198-5203 (2008).
4. Nair, A.B. & Jacob, S. A simple practice guide for dose conversion between animals and human. *J Basic Clin Pharm* **7**, 27-31 (2016).
5. Tang, H., Shao, H., Yu, C. & Hou, J. Mcl-1 downregulation by YM155 contributes to its synergistic anti-tumor activities with ABT-263. *Biochemical pharmacology* **82**, 1066-1072 (2011).
6. Na, Y.S., *et al.* YM155 induces EGFR suppression in pancreatic cancer cells. *PLoS One* **7**, e38625 (2012).

REVIEWERS' COMMENTS:

Reviewer #4 (Remarks to the Author):

All points raised in the previous round of review have been adequately addressed. No further comments from me.